# Development of peptides for targeting cell ablation agents concurrently to the Sertoli and Leydig cell populations of the testes: An approach to non-surgical sterilization

Barbara Fraser[1,2,3], Alex Wilkins[1,2,3], Sara Whiting[1,2,3], Mingtao Liang[4,5], Diane Rebourcet[1,2,3], Brett Nixon[1,2,3], Robert John Aitken[1,2,3]*

1 Priority Research Centre for Reproductive Science, University of Newcastle, Callaghan, NSW, Australia, 2 Pregnancy and Reproduction Program, Hunter Medical Research Institute, New Lambton Heights, NSW, Australia, 3 College of Engineering, Science and Environment, University of Newcastle, Callaghan, NSW, Australia, 4 School of Biomedical Sciences and Pharmacy, University of Newcastle, Callaghan, NSW, Australia, 5 College of Health, Medicine and Wellbeing, University of Newcastle, Callaghan, NSW, Australia

* John.aitken@newcastle.edu.au

**Data Availability Statement:** All relevant data are publicly available from the University of Newcastle

## Abstract

The surgical sterilization of cats and dogs has been used to prevent their unwanted breeding for decades. However, this is an expensive and invasive procedure, and often impractical in wider contexts, for example the control of feral populations. A sterilization agent that could be administered in a single injection, would not only eliminate the risks imposed by surgery but also be a much more cost-effective solution to this worldwide problem. In this study, we sought to develop a targeting peptide that would selectively bind to Leydig cells of the testes. Subsequently, after covalently attaching a cell ablation agent, Auristatin, to this peptide we aimed to apply this conjugated product (LH2Auristatin) to adult male mice *in vivo*, both alone and together with a previously developed Sertoli cell targeting peptide (FSH2Menadione). The application of LH2Auristatin alone resulted in an increase in sperm DNA damage, reduced mean testes weights and mean seminiferous tubule size, along with extensive germ cell apoptosis and a reduction in litter sizes. Together with FSH2Menadione there was also an increase in embryo resorptions. These promising results were observed in around a third of all treated animals. Given this variability, we discuss how these reagents might be modified in order to increase target cell ablation and improve their efficacy as sterilization agents.

## Introduction

Dogs and cats are the most common domesticated carnivores across the world [1]. Estimates of the world dog population vary widely but is generally believed to be upwards of between 500–600 million [2,3]; approximately 75% (~480 million animals) of which are considered to be free-ranging or strays [4]. Additionally, the feral cat population is estimated to be at least 100 million, including 60 million in the USA and up to 12 million in Australia alone [5, 6]. These numbers highlight the magnitude of the worldwide problem posed by free-roaming

NOVA repository (https://doi.org/10.25817/h7wg-t184).

**Funding:** RJA was in receipt of this award The grant number was D1213-W16 The funder was the Michelson Found Animals Foundation The URL is https://www.foundanimals.org/ No The funders had no role in study design, data collection and analysis, decision to publish, or preparation of the manuscript.

**Competing interests:** The authors have declared that no competing interests exist.

domestic dogs and cats, which impacts not only human health but also imposes significant economic and environmental costs.

Traditional means of lethal control used to mitigate the impact of free-roaming populations of pest species are increasingly becoming unacceptable to communities, non-governmental agencies and welfare organizations because of their perceived lack of humaneness [7]. Additionally, there remain questions about the efficacy of such strategies and, particularly in the case of poisonous baiting, the impact of the toxicants on non-target species and the environment at large [7]. Surgical sterilization is one alternative to pest species management that is viewed as both effective [8] and more socially acceptable than culling. However, relative to an injection, surgical sterilization is labor intensive and expensive and necessitates the use of anaesthesia, experienced personnel, and specialized facilities. An ideal alternative to lethal control or surgical sterilization would be to develop a safe effective means of non-surgically sterilizing pest animal species.

Indeed, nonsurgical fertility control is increasingly advocated as both a more humane [9, 10] and cost-effective [7] solution for free-roaming dog and cat management and to resolve human-wildlife conflicts. Additionally, nonsurgical sterilization may enable a higher throughput, compared to surgical sterilization. Currently-used, non-surgical sterilization technologies include: (i) hormonal methods such as implants [11]; (ii) the induction of immunocontraception that relies on antibody production against reproductive cells or signalling molecules; and (iii) reproductive senescence induced by a chemosterilant [12, 13]. Hormonal and immuno-contraception require periodic re-administration which may not always be practicable, while major disadvantages of untargeted chemosterilants are that they are often not species specific, require anaesthesia and highly trained administrants [12, 14], may have off-target effects at the doses required for sterilization [15], and/or may be environmental pollutants. Other than intratesticular injection of the aforementioned chemosterilants, to date there have been no injectable agents developed, that achieve a permanent state of sterilization.

To address this problem, we have initially focused on the male and sought to develop principles by which targeting peptides could be designed to selectively deliver cell ablation agents to the testis and thereby disrupt the process of spermatogenesis. Ideally, these agents could be administered subcutaneously and would only have to be administered once to effect complete sterilization of the host. Specifically, our aim was to harness the mouse model to develop targeting vectors for two specialized cell populations; the Sertoli cells that nurse differentiating male germ cells and form the protective blood-testis-barrier [16], and the androgen producing Leydig cells, that reside within the interstitial space [17, 18]. Our approach builds on previous evidence that peptides have utility for selectively delivering cytotoxic agents to the gonadotrophic cells of the pituitary [19, 20] or the Sertoli cells of the testes [21,22], whereupon they transiently disrupt male fertility. To extend this response to achieve a state of permanent sterility, we elected to target cytotoxic agents to the Sertoli and Leydig cell populations concurrently, with the ultimate goal of ablating a sufficient number of these cells to permanently disrupt spermatogenesis. Sertoli and Leydig cells are regulated by Follicle-Stimulating Hormone (FSH) and Luteinizing Hormone (LH) respectively, through their association with gonadotrophin specific receptors at the cell surface. Fortuitously, the unique expression of FSH receptors by Sertoli cells and LH receptors by Leydig cells, facilitates specific targeting of agents to these cell populations, thus mitigating the risk posed by off-target effects.

This strategy builds on previous proof-of-concept studies in which we have developed FSH peptides (FSH2) that exhibit selective binding capability and specificity for the TM4 Sertoli cell line *in vitro* and home to the testes following *in vivo* administration. Moreover, the conjugation of FSH2 to the redox cycling agent, 2-methyl-1,4-naphthoquinone (also known as menadione or Vitamin K3) proved effective in eliciting persistent oxidative damage within the male

germ cells such that the spermatozoa of treated mice exhibited significant DNA damage, however it was not sufficient to compromise male fertility [22]. Clearly the sterilizing effects we were seeking, could not be achieved by targeting the Sertoli cells in isolation.

Hence, to augment this response, here we have designed an additional peptide delivery vector based on the amino acid residues of the LH β-subunit; corresponding to the equivalent domain incorporated into the FSH2 peptide. FSH, LH, Thyroid-Stimulating Hormone (TSH), and human Chorionic Gonadotropin (hCG) are all glycoprotein hormones [23] that consist of an α-subunit, common to all, and a β-subunit that confers the specificity for binding to the cognate receptor [23]. There is, moreover, a common binding mode between the glycoprotein hormone β-subunits and their receptors [23]. In this study, a Leydig cell-targeting peptide was covalently attached to the anticancer cell ablation agent, Auristatin, prior to *in vivo* co-administration with FSH2Menadione. The purpose of this study was to examine the ability of an Auristatin-LH peptide construct to disrupt Leydig cell function. Additionally, we wished to determine whether the inhibition of Leydig cell function by the Auristatin-LH peptide might act synergistically with the induction of enhanced ROS generation by the Sertoli cell population using the FSH2Menadione, to achieve the permanent disruption of fertility. If this outcome could be achieved, it would provide the basis for an injectable, accessible, and cost-effective method of nonsurgical sterilization.

## Materials and methods

### Materials

Unless otherwise stated, chemicals and reagents were purchased from either Merck Darmstadt, Germany) or Thermo Fisher Scientific (Waltham, MA, USA). Hydrochloric acid (HCl; 32% w/v AnalaR) was supplied by VWR International (Radnor, PA, USA). Protein Lo-Bind Eppendorf tubes (1.5 ml) were supplied by Eppendorf (AG Hamburg, Germany). Sodium chloride (NaCl), and tris(hydroxymethyl)aminomethane (Tris) were purchased from Astral Scientific, (Sydney, NSW, Australia). Fetal bovine serum was purchased from Interpath (Heidelberg West, Vic, Australia). All peptides were purchased from GenScript (Piscataway, NJ, USA) with C-terminal amidation and more than 98% purity. All cell lines were purchased from ATCC (Manassas, Virginia, USA) except the mECap18 cells that were obtained as a generous gift from Dr Petra Sipilä (Institute of Biomedicine, Department of Physiology, University of Turku, Turku, Finland). Bioss rabbit polyclonal anti-LH receptor antibody (bs-6431R) and donkey serum were purchased from Sapphire Bioscience (Redfern, NSW, Australia). Goat polyclonal anti-8-hydroxydeoxyguanosine (8-OHdG) was purchased from Millipore (Temecula, CA, USA). Agarose (low melting point) was obtained from Applichem GMBH (64291 Darmstadt, Germany) and ethanol purchased from POCD Scientific (North Rocks, NSW, Australia). DNA/RNA Damage Antibody (15A3) was obtained from Novus Biologicals (Littleton, CO, USA). Maleimide-vcMMAE (Auristatin) was purchased from MedChemExpress (Monmouth Junction, NJ, USA).

### Design of LH receptor targeting peptide

To enable specific targeting of Leydig cells, a peptide was designed that possessed high binding capability and specificity as well as the *in vivo* stability required for the peptide to reach its target. We have previously developed a peptide based on the residues of the β-subunit of FSH that specifically targets Sertoli cells [22]. Here, we employed the same design principles to design a peptide corresponding to residues occupying the equivalent "seatbelt" region of the mouse LH β-subunit; a region known to be important for gonadotrophin receptor binding [24–27]. An important modification of the native sequence was that all cysteine residues,

except those that occurred on the end of the molecule, were substituted with serine residues. This modification most closely mimics the native 3-D structure of the peptide, yet reduces the propensity for peptide cyclization to occur. Moreover, removal of internal reactive thiol groups ensured a single site was available for maleimide-thiol mediated coupling of the peptide to the desired payload and thus avoided the potential for mixed end products to be formed. Finally, the C-terminal was amidated to inhibit protease degradation of the peptide *in vivo*.

### Physicochemical properties and structure of LH peptides

A comparison was made of the physicochemical properties of LH synthetic peptides using data obtained from PepCalc (PepCalc.com, 2015 Innovagen AB). Furthermore, Swiss-Prot Deep Viewer software was used to map each of the peptides onto the core structure of LHβ Swiss-Model O09108 (LSHB_MOUSE) *Mus musculus (Mouse)* Lutropin subunit beta Template: 1hcn.2.B *"HUMAN CHORIONIC GONADOTROPIN" P0DN86*SMTL Version: 2019-12-06, Seq Identity: 63.96%, Seq Similarity: 0.52.

### Cell culture

All cell lines were maintained in a humidified atmosphere at 37˚C and 5% $CO_2$. MLTC1 Leydig cells were cultured in growth medium composed of RPMI 1640 medium supplemented with 10% (v/v) fetal bovine serum, and 100 U/mL penicillin-streptomycin. TM3 Leydig cells were cultured in growth medium composed of F12/Dulbecco's modified Eagle's medium, 2.5 mM L-glutamine, 0.5 mM sodium pyruvate, 15 mM HEPES, 2.5% of a mixture of 92.5% horse serum and 5% fetal bovine serum, and 100 U/mL penicillin-streptomycin. MLTC1 and TM3 cells were sub-cultured at a ratio of 1:5 into cell culture flasks. Immortalized mouse proximal caput epididymal epithelial (mECap18) cultures [28] were used as a nonspecific cell control for investigating the specificity of LH peptide interactions and were maintained in growth medium composed of DMEM supplemented with 10% (v/v) fetal bovine serum, 2 mM L-glutamine, 100 U/mL penicillin-streptomycin, 1 mM sodium pyruvate and 50 nM 5-α-androstan-17β-ol-3-one and sub-cultured at a ratio of 1:10 into culture flasks. Cells were grown to 80% confluency before being harvested using 0.5% trypsin-EDTA, quenched, centrifuged at $300 \times g$ for 5 min, and then washed in warmed growth medium. After centrifugation, the cells were resuspended in warm medium and subjected to incubation with peptide- fluorescein isothiocyanate (FITC) conjugates.

### Animals and animal care, injections and monitoring

In this study, adult (>8 week old) male Swiss (CD1) mice were obtained from a breeding colony at the University of Newcastle Central Animal House and housed under a conventional controlled light and temperature regimen (12-h light: 12-h dark cycle, 21–22˚C). All procedures involving mice were conducted in accordance with the Guide for the Care and Use of Laboratory Animals as adopted and promulgated by the U.S. National Institutes of Health. Additionally, mice were handled, monitored and euthanized with the approval of the University of Newcastle's Animal Care and Ethics Committee (approval number A-2014-417) in accordance with NSW Animal Research Act 1998, NSW Animal Research Regulation 2010 and the Australian Code for the Care and Use of Animals for Scientific Purposes 8th Ed.

### Validation of the MLTC1 Leydig cell model

To ascertain the binding capability of synthetic LH peptides, we required a Leydig cell line that expressed the LH receptor. The two mouse Leydig cell lines, TM3 and MLTC1, are both

expected to express the LH receptor. RNA was extracted from testis, MLTC1 Leydig cells, TM3 Leydig cells and the mouse epididymal epithelial (mECap18) cells, with the latter serving as a negative control. To determine which of the two Leydig cell lines had the highest relative LHr expression quantitative PCR (qPCR) was used to assess expression relative to the housekeeping gene, GAPDH. Subsequently, LHr protein expression was assessed in cultured cells using anti-LHr antibodies.

## RNA extraction, reverse transcription and quantitative PCR (qPCR)

Total RNA isolation was performed using two rounds of a modified acid guanidinium thiocyanate–phenol–chloroform protocol [29] followed by isopropanol precipitation. Total RNA was DNase treated prior to reverse transcription to remove genomic DNA. Reverse transcription to cDNA was conducted using M-MLV Reverse Transcriptase (Promega, M1701) following the manufacturer's instructions with subsequent confirmation of transcription using cyclophilin qPCR (cyclophilin primers (5′-CGTCTCCTTCGAGCTGTTT-3′; 5′-ACCCTGGCACATG AATCCT-3′); $T_m$ 61°C).

Quantitative PCR: Expression of LHr relative to, the housekeeping gene, GAPDH, was conducted as follows: Taq qPCR was performed using SYBR Green GoTaq qPCR master mix (Promega) according to manufacturer's instructions on a LightCycler 96 SW 1.0 (Roche Diagnostics GmbH, Mannheim, Germany) thermocycler. Primer sequences for the LH receptor and GAPDH have been supplied as supplementary data (S1 Fig). 20 µl reactions were performed on cDNA equivalent to 100 ng of total RNA and conducted for 45 amplification cycles. SYBR Green fluorescence was measured after the extension step at the end of each amplification cycle and quantified using LightCycler Analysis Software (Roche). For each sample, replicates omitting the reverse transcription step were undertaken as negative control. Each PCR was performed on at least 3 separate cell or tissue isolations, of which a representative average is shown. $C_t$ values less than 30 were assumed to be primer dimers and were treated as negative for expression. Relative expression was calculated using the $\Delta\Delta C_t$ method.

## Validation of MLTC1 Leydig cells

Immunocytochemistry was conducted to confirm expression of the LH receptor protein in MLTC1 Leydig cells using mECap18 cells as a negative control. Cells were grown to 80% confluence on sterile poly-L-lysine-coated round coverslips within the wells of 24-well culture plates. Cells were fixed *in situ* with 4% paraformaldehyde (PFA) for 10 min at room temperature (RT). After rinsing with buffer 1 [phosphate buffered saline (PBS) with 3% BSA and 0.015% Triton X-100], cells were blocked for 1 h at RT with buffer 2 (PBS with 10% goat serum, 3% BSA and 0.015% Triton X-100) and then incubated overnight at 4°C with Bioss rabbit polyclonal anti-LHr (diluted to 4 µg/ml in buffer 1). Cells were then re-blocked in buffer 2 for 15 min at RT before incubating for 1 h at RT in 1/200 of goat anti-rabbit AF488 in buffer 2. Finally, cells were washed with buffer 3 (PBS plus 0.015% Triton X-100) then counterstained using a Far Red nuclear stain (diluted 1/2000 in PBS) for 30 min at RT. After washing in PBS, coverslips were removed from the wells and inverted onto a drop of Mowiol on glass microscope slides then sealed using clear nail varnish before cells were imaged using fluorescence microscopy.

## Assessment of in vitro peptide binding capability and specificity

MLTC1 cells and mECap18 cells were used to assess the relative binding capabilities and the specificity of targeting of the LH peptides. All peptides (Table 1) were purchased with an N-terminal spacer, 1,6-aminohexanoic acid (Ahx), attached to FITC and dissolved in DMSO to

form a 2.5 mM stock solution. Assessment of binding capability was conducted as previously described [22]. Briefly: Cells were grown to 80% confluency before harvesting and treatment with the LH-FITC in concentrations ranging from 0 to 50 μM. The cells were incubated at 37˚C for 1 h. Cells were then stained with Live/dead Live/dead Far Red stain incubated at 37˚C for 20 min before washing and resuspension in HBSS. FITC staining and was then analyzed using fluorescence assisted cell sorting (FACS) as previously described [22]. The median fluorescence intensity was recorded for each treatment group and the shift in median fluorescence intensity was then calculated by subtracting the median fluorescence intensity of the vehicle (0 μM) from the median fluorescence intensity of each dose.

## Assessment of *in vivo* targeting and off-target binding by LH peptides

All LH peptides were assessed *in vivo* to determine the efficacy of targeting the testes and the extent of off-target binding. A total of 300 μL of 1 mM peptide-FITC or the vehicle (30% Kolliphor/PBS) was injected intraperitoneally (IP) into adult male Swiss CD1 mice, with three males being randomly assigned to each treatment group. Mice were euthanized 8 h post-injection and tissues (adrenals, brain, epididymides, kidneys, liver, lungs, pituitary, seminal vesicles, spleen, and testes) were collected, weighed and immediately imaged. Tissues were imaged in cold PBS with a Leica MZFLIII microscope with an epifluorescent green filter. All images included the vehicle control to enable comparison of median fluorescence intensity of the treatments relative to the control using ImageJ software [ImageJ version 1.52a; Wayne Rasband, National Institutes of Health, USA. http://imagej.nih.gov.ij, Java 1.8.0_112 (64-bit)].

## Conjugation of Auristatin to LH peptide

LHa peptide (Table 1) was obtained from GenScript with a γ-aminobutanoate-mercaptopropionic acid linker (hereinafter referred to as LH2 peptide) and was attached, via the N-terminal thiol, to maleimide-Auristatin (in the form of maleimide-valine-citrulline- para-aminobenzyloxycarbonyl -monomethyl-Auristatin E) as follows. LH2 peptide (MW 1333) was dissolved in 10 mM PBS, pH 7, at a concentration of 8.5 mg/250 μL. The peptide was then reduced, using immobilized TCEP (tris(2-carboxyethyl)phosphine) disulphide reducing gel, the solution pH adjusted to 7, and then deoxygenated. Meanwhile, 1.2 molar equivalents of maleimide-Auristatin (MW 1316) were dissolved in 53% dimethyl formamide/10 mM PBS, pH 7, at 10mg/3 mL and placed in a reaction vessel. The maleimide-Auristatin solution was then deoxygenated, and the reduced peptide solution was added dropwise, using a syringe, whilst stirring and under positive pressure with nitrogen gas. The reaction was then stirred overnight at RT. The success of the conjugation reaction was confirmed by mass spectrometry, using a Bruker Ultraflextreme MALDI-TOF/TOF (mode: reflectron positive; matrix: α-cyano-hydroxy-cinnamic acid). The solvents were removed from the reaction by freeze drying under vacuum. The dried reaction mix was then reconstituted and purified using reverse phase HPLC.

**Table 1. Amino acid sequences and properties of the unmodified LH receptor binding peptides.**

| Code | Position on LHβ | Amino acid residues | Predicted aqueous solubility | Charge at pH7 | Ratio of hydrophobic/ hydrophilic residues |
|------|-----------------|---------------------|------------------------------|---------------|--------------------------------------------|
| L57 | LHβ 38–57 | CVRVLPAALPPVPQP-NH$_2$ | poor | +1.9 | 4 |
| L95 | LHβ 81–95 | CSFPVALSSRSGPSRR-NH$_2$ | good | +3.9 | 0.6 |
| L101 | LHβ 93–101 | CRLSSSDCG-NH$_2$ | good | +0.9 | 0.6 |
| LHa | LHβ 96–107 | SSSDAGGPRTQP-NH$_2$ | good | +1 | 0.3 |

Physicochemical properties of LH synthetic peptides calculated using PepCalc.

Samples were eluted from the Waters 250 × 10.0 mm Jupiter 4u Proteo 90A column using a gradient method with Solvent A (0.1% v/v formic acid in $H_2O$) and Solvent B (90% acetonitrile with 0.1% v/v formic acid) with a flow rate of 2 mL/min over 37 min followed by 2 min at 100% Solvent B before re-equilibration for 5 min. The eluted species were detected by their absorbance at 220 nm. Peak fractions were collected and LH2-Auristatin identity and purity was verified using Orbitrap liquid chromatography/mass spectrometry before the product was freeze-dried under vacuum and stored as a powder at -20˚C.

## Conjugation of FSH2 peptide with menadione

FSH peptides were attached, via the N-terminal thiol, to menadione as previously described [22]. Briefly, deoxygenated aqueous solution of reduced FSH2 peptide was added dropwise to 10 molar equivalents of menadione in DMSO in a sealed reaction vessel, heated in a water bath to 55˚C, whilst stirring with a magnetic bar and under positive pressure of nitrogen gas. After peptide addition to the menadione suspension, stirring of the reaction was continued at 55˚C for 2 h. Mass spectrometry was used to confirm the success of the conjugation reaction, using a Bruker Ultraflextreme MALDI-TOF/TOF as above. The reaction product was dried and then purified using reverse phase HPLC. Peak fractions were collected and peptide-quinone identity and purity was verified using MALDI-TOF/TOF mass spectrometry, as above, before the product was freeze dried under vacuum and stored as a powder at -20˚C.

## *In vivo* pilot study FSH2Menadione with LH2Auristatin

Four treatment groups of ten 55-day-old male Swiss male CD1 mice were injected intraperitoneally. Treatments included: (1) 00 μL vehicle control (30% Solutol (Kolliphor)/PBS), (2) 300 μL/30 g 14.5 mM FSH2Menadione, (3) 100 μL/30g 425 μM LH2Aur, or (4) 300 μL/30 g 14.5 mM FSH2Menadione plus 100 μL/30g 425 μM LH2Aur (consecutive injections, 16 h apart). The dose of 1 mg/kg of conjugated Auristatin was determined with reference to the work by Doronina et al. [30], who determined the maximum tolerated dose (MTD) of conjugated Auristatin for mice is 1.1 mg/kg. Solubility of the FSH2Menadione was limited to 14.5 mM (25 mg/kg) in the biocompatible reagents (Kolliphor/PBS) delivering a dose much lower than menadione's MTD of 830 mg/kg. Six weeks post injection, five randomly selected mice from each treatment group were mated with two untreated females (10–12 weeks old) each for three consecutive weeks. Females were monitored for mating plugs each day and on the thirteenth day post-mating plug, or if weight gain indicated approximately 13 days of pregnancy, females were euthanized weighed, embryos and resorption moles were counted, ovaries were collected, weighed, and corpora lutea counted. After the mating period, at ten weeks post-injection, all males were euthanized, weighed and blood collected by cardiac puncture. Blood samples from each animal were decanted into EDTA-coated tubes and plasma was separated (10 min, 400 g). Plasma was stored at -80˚C for later measurement of serum FSH levels, for which a AVIVA FSH ELISA Kit was purchased from Sapphire Bioscience (Redfern, NSW, Australia) and conducted as per manufacturer's instructions.

Tissues (seminal vesicles, epididymides, adrenals, kidneys, liver, brain, heart, spleen and testes) were collected and weighed. A sample of each tissue was fixed in Bouin's solution (75 mL picric acid, 40% aqueous solution; 25 mL formalin, 40% aqueous solution; and 5 mL glacial acetic acid) for around 40 min per mg tissue. One testis from each male was fixed and the other snap frozen in liquid nitrogen and stored at -80˚C. After fixing, testes were washed with 70% ethanol until the solution was clear and then processed for paraffin embedding, sectioning (5 μm thickness) and haematoxylin and eosin staining (H and E). H and E-stained sections were used to evaluate cross-sectional tubule areas as described below. To evaluate the impact

on Sertoli cells numbers, an anti-SOX9 antibody was used to stain sections, while an ApopTag Kit (Merck) was used to detect apoptotic cells in accordance with the manufacturer's instructions.

Immediately after dissection, epididymal spermatozoa were also collected for analysis of motility, vitality, DNA strand breaks (HALO), DNA fragmentation (SCSA), the presence of reactive oxygen species (MSR and DHE) and oxidative DNA adducts (anti-8-OH-dG). For these assays, three incisions were made in the cauda epididymides with a surgical blade before placing the tissue in prewarmed Biggers-Whitten-Whittingham (BWW) medium [31] with 5 U/ml penicillin, and 5 mg/mL streptomycin, pH 7.4, at 37˚C. Spermatozoa were allowed to swim-out for 10 min before being collected into clean tubes. Sperm motility was assessed using a HTM-IVOS II Computer Assisted Sperm Analysis and sperm vitality was assessed using the eosin exclusion test [32] and viewed using a light microscope at 40× magnification. At least 100 spermatozoa per sample were analysed to determine percentage vitality of each sample. Spermatozoa were stained for FACS analysis of mitochondrial and cytosolic superoxide generation using MSR and DHE staining [33, 34]. Additionally, sperm smears were made for staining with Rapid Diff (Pathtech, Preston, Vic, Australia), as per manufacturer's instructions, to evaluate sperm morphology. Analyses of DNA strand breaks (HALO), and detection of the oxidative DNA adducts (anti-8-OH-dG) were conducted as previously described [22]. At least 100 sperm/sample (10,000 sperm/sample for FACS) were assessed for each assay and, where possible, all counts were conducted blind. The remaining spermatozoa were aliquoted into tubes containing $10^6$ sperm/mL, snap frozen in liquid nitrogen and stored at -80˚C until use. Thawed sperm cells were later assessed using FACS for DNA fragmentation using a sperm chromatin structure assay (SCSA) as previously described [35, 36]. All FACS data were acquired and analyzed using BD FACSDiva software (BD Biosciences), with a total of 10,000 events collected per sample. Only results for viable cells are reported for all FACS assays.

## Cross-sectional area of tubules

To ascertain average seminiferous tubule area, six H and E-stained testes sections were randomly chosen from each treatment group. Sections were imaged using a 10 × objective under bright field microscopy and then images (4080 × 3072 pixels) were imported into ImageJ software. At least 50 tubules from each testis section were outlined and the area in pixels/tubule calculated.

## SOX9 staining to ascertain Sertoli cell number

Testes sections (5μm thick) were dewaxed using xylene and then rehydrated in an ethanol gradient. Heat-assisted epitope retrieval was conducted in Tris-EDTA, pH 9. Slides were then incubated in 0.3% hydrogen peroxide in Tris buffered saline (TBS) for 30 min at RT. After washing in TBS (3 × 5 min), the sections were blocked using 5% normal goat serum (NGS) for 30 min at RT in a humidified chamber. The sections were then incubated with rabbit anti-SOX9 antibodies (Merck: AB5535) in NGS (1:500) overnight at 4˚C. Slides were washed in TBS (3 x 5 min) prior to incubation with biotinylated goat anti-rabbit antibodies (Abcam, ab6720) in NGS (1:200) for 30 min in a humidified chamber at RT. Slides were washed in TBS (3 × 5 min) then incubated with streptavidin-HRP (Abacus SA-5004) in TBS (1:1000) for 30 min at RT before being washed again in TBS (3 × 5 min). Sections were then treated with 3,3'-diaminobenzidine (DAB) for up to 5 min or until sufficient colour development. Slides were washed with TBS (3 × 5 min) and counterstained with haematoxylin for 8 min before being dehydrated and mounted with Eukitt. Slides were imaged using bright field microscopy and the number of Sertoli cells counted in all tubules across one section from each slide.

## Statistical analysis

All experiments were conducted at least three times on individual samples in technical triplicate. All quantitative data was expressed as the mean ± standard error from three or more replicate samples. Statistical analysis of data was conducted using Microsoft Excel (v16.0; Microsoft Corporation, Redmond, WA, USA) as follows: a single factor ANOVA was conducted on the data from all treatments. If the p-value was < 0.05, then pair-wise Student's t-tests were conducted using two-samples assuming unequal variance if difference between the variances was > 2-fold, otherwise assuming equal variance.

## Results

### Design of LH receptor binding peptides

A LH receptor binding peptide (hereinafter referred to as LHa) was designed based on equivalent residues to those incorporated into a previously developed FSH peptide mimic [22]. Specifically, the LHa peptide incorporated amino acid residues mapping to LH β-subunit 96–107 (Table 1). An additional three LH peptides were also designed based on sequences that confer specific receptor binding. These were designated L57, L95 and L101 depending on where their sequence mapped within the LH β-subunit. L57 peptide included the amino acid residues LHβ 38–57 [26], L95 peptide the residues in LHβ 81–95 [25] and L101 peptide included the residues LHβ 93–101 [23] (Table 1).

### All LH peptides mapped to the surface of the receptor binding regions of LH β-subunit

The position of the peptides mapped to surface of the 3-D structure of the LH β-subunit (Fig 1) confirmed that the new mouse amino acid sequences were located in the βL2 loop and the "seatbelt", which are the two binding regions on the surface of the β-subunit (Fig 1). Additionally, their position on the surface of the protein means that they are accessible for receptor binding. The positioning of the peptides on the surface of the protein within the recognised binding regions was expected to confer a high likelihood of binding to the LH receptor.

A comparison of the physicochemical properties of the peptides (Table 1) predicted good aqueous solubility for all of the peptides except L57. This may be expected as the ratio of hydrophobic to hydrophilic residues for this peptide was four, considerably higher than that of the other, more soluble, peptides which all had an equivalent ratio of less than one. The predicted charge at neutral pH on LHa and L101 were both around +1. L57 was predicted to have a higher overall charge of +1.9, whereas L95 was predicted to possess a very high +3.9 charge at neutral pH. In recognition that both hydrophobicity and charge can influence the affinity of a peptide for the cell surface (with positive charges increasing the electrostatic attraction for negative cell surfaces), we elected to directly assess the binding specificity of each peptide (see section entitled '*LH peptides demonstrated variable in vitro binding capability and specificity*').

### Q-PCR and immunocytochemistry confirmed MLTC1 Leydig cells as an in vitro model

For this study, as we aimed to target the LH receptor, we required a cell line that displayed strong expression of the LH receptor gene (*Lhcgr*) to act as an *in vitro* model. qPCR and an anti-LHr antibody were used to probe for the presence of the LH receptor in both MLTC1 and TM3 Leydig cell lines, as well as in the control epididymal epithelia cell line (mECap18). Using qPCR, we confirmed that of the two available mouse Leydig cell lines, the MLTC1 Leydig cell line expressed the *Lhcgr* gene (S2 Fig). By contrast, no *Lhcgr* transcripts were detected in the

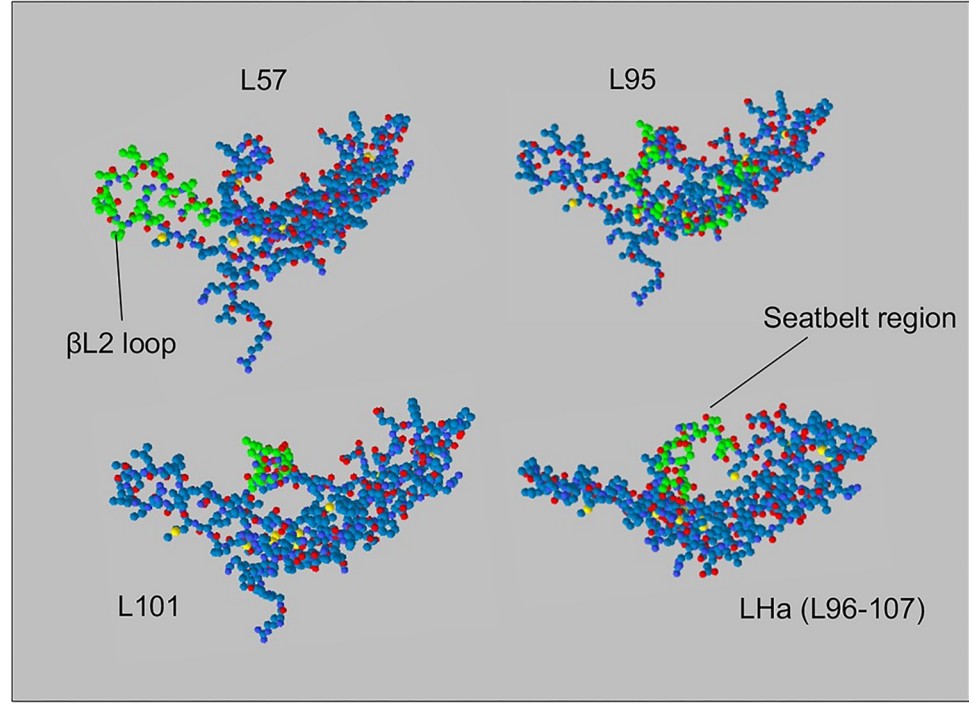

**Fig 1. Mapping the peptides to the β- subunit of the LH protein which targets receptors on theca cells in the ovary and Leydig cells of the testis.** LHa rationally designed based on binding residues of β-subunit of FSH protein with FSH receptor [37]. Swiss-Prot Deep Viewer was used to map the peptides (in green) to the surface of **Swiss-Model O09108 (LSHB_MOUSE)** *Mus musculus (Mouse)* **Lutropin subunit beta** Template: **1hcn.2.B** *"HUMAN CHORIONIC GONADOTROPIN" P0DN86*SMTL Version: 2019-12-06, Seq Identity: 63.96%, Seq Similarity: 0.52. The βL2 loop and the seatbelt region are indicated by arrows and are the receptor-binding regions of the β-subunit of the glycoprotein hormones.

TM3 cell line or in the negative control mECap18 cell line. These data were validated by immunocytochemistry, which confirmed LH receptor expression within the MLTC1 Leydig cells, but not in mECap18 cells (S2B Fig). Given these data we elected to use MLTC1 cells as the *in vitro* Leydig cell line model.

## LH peptides demonstrated variable in vitro binding capability and specificity

MLTC1 cells and mECap18 cells were used to assess the relative binding capabilities and the specificity of targeting of synthetic LH peptides, revealing that each peptide bound to the MLTC1 Leydig cells in a dose-dependent manner. This was evidenced by a proportional shift in the median fluorescence intensity of the stained MLTC1 cell population across the dose range of applied peptides (Fig 2A). Additionally, each of the peptides displayed suitable specificity with three-fold higher median fluorescence intensity when applied to MLTC1 Leydig cells as compared to the mECap18 control cells (Fig 2B and 2C). The peptide with the greatest shift in median fluorescence and, by inference, the greatest binding capability, was L95 (Fig 2B); which possessed more than double the MLTC1 cell binding affinity than that of the other peptides tested. However, L95-FITC showed a lack of specificity at the highest dose tested, 50 μM, with this peptide exhibiting only 1.5-fold relative binding on MLTC1 cells above that of the background mECap18 cell binding.

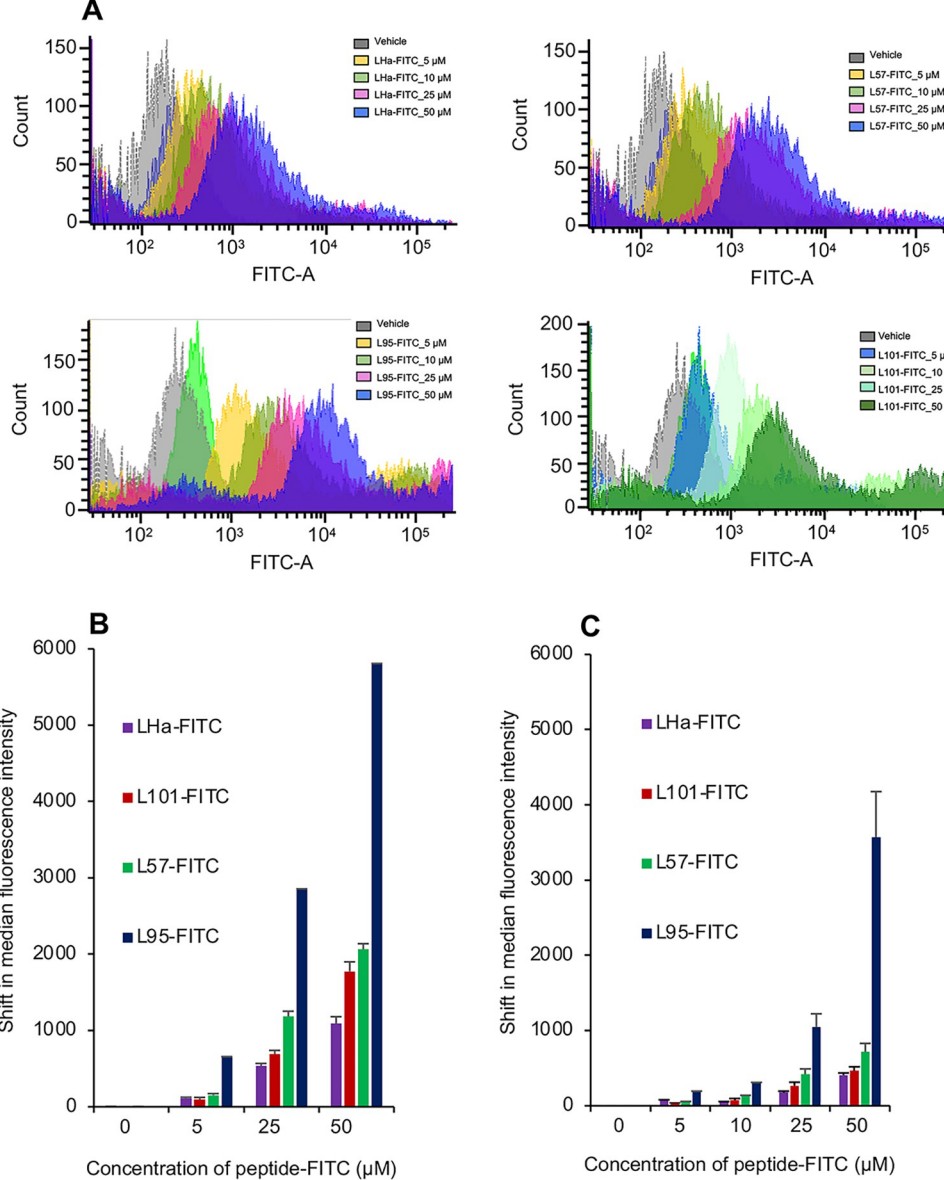

**Fig 2. Peptide targeting and binding affinity *in vitro*.** Peptide-FITC in DMSO or DMSO was applied 1 in 100 parts (v/v) to the cells to make final concentrations of 0 to 50 μM for 1 hour at 37˚C. **A**. FACS analysis of binding of FITC-conjugated LH peptides to MLTC1 Leydig cells (n = 3) using shift in median FITC fluorescence intensity in response to increasing concentration. **B.** Comparison of binding of LH-peptides to MLTC1 Leydig cells (n = 3), and **C.** mEcap18 cells. FACS analysis of binding of the peptides using shift in median fluorescence intensity (n = 3).

## LH peptides localized to the testes *in vivo*

Given the promising *in vitro* results, we sought to verify the homing potential of the LH-FITC conjugates to their testicular target following injection into adult male mice. We failed to detect any significant differences in the weight of mice exposed to either the vehicle or the peptide-FITC treatments eight hours post-injection. Similarly, we did not record any differences in the weight or gross morphology of the representative tissues collected. All four LH peptides exhibited localization within the testes with a clear difference in the fluorescence intensity being detected in the testes of the vehicle-injected controls, compared to that of the peptide-

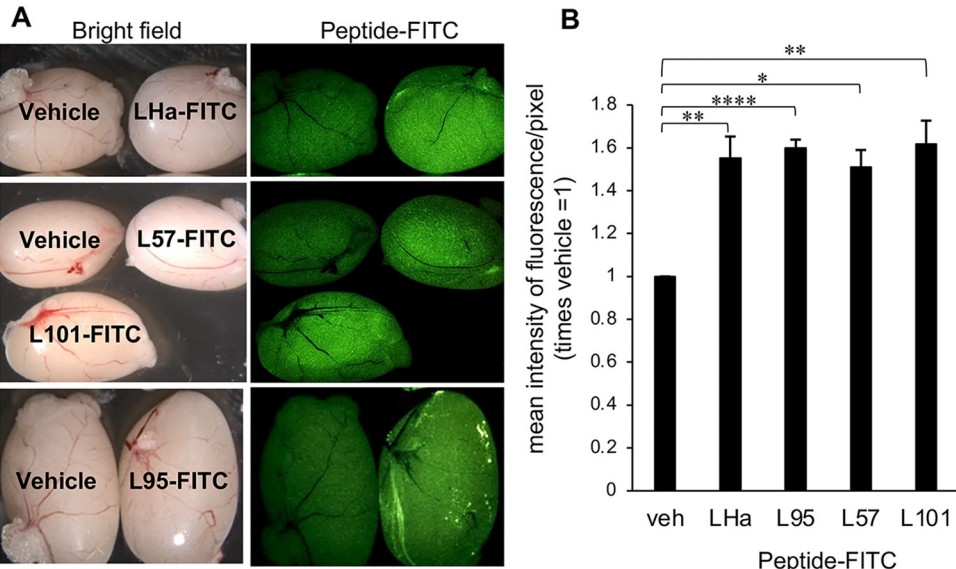

**Fig 3. Confirmation of *in vivo* targeting by LH peptides.** Representative images of peptide-FITC localised to testes 8 hours post-injection. Male adult mice were injected IP with 300 μl of either 1mM peptide-FITC in 30% Kolliphor/PBS or 30% Kolliphor/PBS with 3 mice in each treatment group. Animals were euthanized 8 hours post-injection and the testes were imaged, using Leica MZFLIII microscope with an epifluorescent green filter, next to the vehicle control. **A.** Representative images of testes after injection with (from top to bottom) LHa-FITC, L57-FITC, L101-FITC and L95-FITC. **B.** Image J analysis of mean fluorescence intensity/pixel in testes of peptide-FITC injected males. Images were analysed using ImageJ software (ImageJ 1.52a, Wayne Rasband, National Institutes of Health, USA. http://imagej.nih.gov.ij], Java 1.8.0_112 (64-bit)). Mean fluorescence intensity/pixel was normalised to that of the vehicle control within the same image. (n = 3). Statistical analysis by One-way ANOVA then post hoc using student T-tests.

FITC injected males (Fig 3A). This contrasted the findings of the adrenals, brain, lungs, pituitary, or seminal vesicles for which no difference in fluorescence staining was observed between the vehicle control and the treated animal tissues. However, there was evidence of clearance of the peptides through the detoxifying and excretory organs of the liver and kidney, both of which presented with relatively low fluorescence labelling (S3 Fig). Additionally, there appeared to be modest FITC fluorescence associated with the spleen in at least a portion of the treated animals, but particularly within the spleens of the L57-FITC treated animals (S3 Fig).

Based on these findings, images of the treated testes were imported into ImageJ to compare the mean fluorescence intensity/pixel. The results were expressed as a ratio of the mean fluorescence intensity/pixel of the treated in any image to the vehicle control in the same image (Fig 3A). All four peptides displayed a similar normalized fluorescence intensity/pixel of between 1.45 and 1.6-fold that of the vehicle control (Fig 3B), albeit with differing levels of significance. These results suggest that the LH peptides possess similar abilities to localize to the testes and that they possess similar stabilities *in vivo*. The presence of the FITC-Ahx tag complexed to the peptide, however, means that important factors that determine the pharmacokinetics, such as the theoretical isoelectric point, the charge and the logP value (a measure of hydrophobicity that increases with increasing hydrophobicity) of the complex differs from that of the unmodified peptide and, most likely, that of the peptide with cargo. Therefore, whilst informative, this method of peptide *in vitro* and *in vivo* tracking may not be definitive, and additional criteria were investigated to aid in our selection of an appropriate LHr targeting peptide.

Among these criteria, aqueous solubility was considered an important factor, both in the context of reaction with Auristatin and for use of biocompatible solvents for *in vivo* delivery.

The Auristatin employed in this study has a valine-citrulline linker (S4 Fig) with a terminal maleimide for ease of conjugation and has very poor aqueous solubility with a logP value of 6.04 (https://www.chemsrc.com/en/cas/646502-53-6_1197872.html). Therefore, so as not to compound this lack of solubility, the targeting peptide needed to possess good aqueous solubility and, as L57 was expected to have poor water solubility (Table 1), it was discounted as an option for this study. Additionally, L95 peptide was considered unsuitable as it exhibited a relatively high positive charge (Table 1) and non-specific binding *in vitro* (Fig 2C); both of which are incompatible with selective delivery of the potent cytotoxic payload to be attached to this peptide. Furthermore, Auristatin was to be conjugated to a γ-aminobutanoate-MPA linker on the N-terminal of the peptide via a thiol-maleimide reaction. The linkers are expected to allow the peptide to bind to the receptor unhampered by the attached Auristatin. Therefore, a cysteine extra to the N-terminal cysteine, such as in L101peptide, means that a non-ideal mixed product of mono-conjugated and bi-conjugated peptide-Auristatin, may result. Receptor binding by a bi-conjugated product may be sterically hindered by the extra Auristatin. Considering all of these factors, and based on its predicted charge, solubility (Table 1), *in vitro* selectivity (Fig 2B and 2C) and similar *in vivo* performance to other assessed peptides (Fig 3B), we selected the LHa peptide as the preferred candidate for delivery of Auristatin to the testes.

## Conjugation of menadione to FSH peptide and Auristatin to LHa peptide

In adapting these sequences for the purpose of carrying a specific payload, we attached a linker, comprised of γ-aminobutanoic acid and mercaptopropionic acid (Linker 2), that provided a reactive thiol group for attaching the peptide to other moieties or to nanoparticles via a maleimide-thiol coupling reaction [38, 39]. Additionally, the C-terminal of all peptides was amidated with the aim of inhibiting protease degradation, increasing stability [40, 41], and thereby prolonging the *in vivo* half-life and extending the time that the peptide is exposed to the target receptor. Additionally, we have previously shown that Linker 2, allows menadione to produce ROS by redox cycling despite its covalent attachment to the peptide [22]. The peptides with linker 2 are hereinafter referred to as LH2 and FSH2. FSH2 was attached, via the N-terminal cysteine, to C3 of menadione and the product, FSH2Menadione, isolated. FSH2Menadione had previously been synthesized, purified and the powdered product stored at -20˚C [22]. LH2 was reacted with maleimide-Auristatin overnight and the product successfully purified by HPLC, freeze dried, and the powdered product stored at -20˚C. The identity and purity of the product was confirmed by LC-MS that confirmed a product of mass ~2648 (representing LHa MW 1,333 + vcMMAE MW 1316). The reagents were resuspended in 30% Kolliphor/PBS immediately before administration to the treatment groups.

## LH2Auristatin homed to the testes and impacted seminiferous tubule morphology and germ cell viability

Treated mice maintained healthy weight gain over a ten-week post-injection surveillance period, with no difference in mean body weights detected between the treatment groups. Additionally, with the exception of the testes (Fig 4A), there was no difference in tissue (seminal vesicles, epididymides, adrenals, kidneys, liver, brain, heart, and spleen) weights or morphology between the treatment groups, indicating that the injected peptides did not elicit pronounced off-target effects. In contrast, it appeared that the testes of at least some of the treated mice had been adversely impacted. Specifically, thirty percent of LH2Auristatin-treated males and an additional twenty percent of FSH2Menadione/LH2Auristatin-treated males presented with at least one testis that was significantly smaller (weights ranging from 41 up to 85 mg) than that of the controls. By comparison, all males that had been treated with either

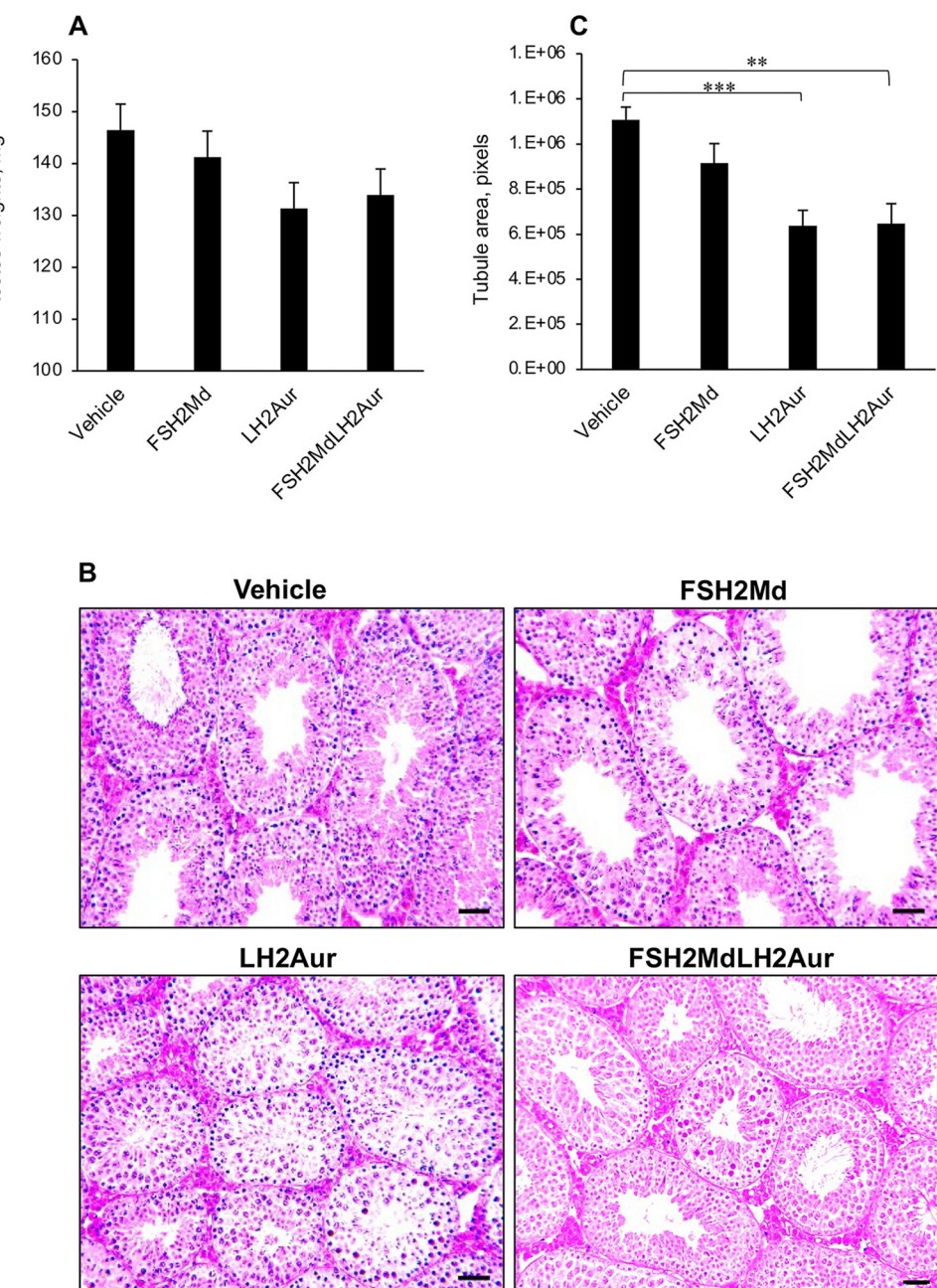

**Fig 4. Effect of FSH2Menadione (FSH2Md) and LH2Auristatin (LH2Aur) *in vivo*.** Male adult mice were injected IP with 300 μl/30 g of either 14.5 mM FSH2Menadione; 100 μl/30 g 420 μM LH2Auristatin; a combination of both, 16 hours apart; or 300 μl/30 g of the vehicle (30% Kolliphor/PBS) with 10 mice in each treatment group. Five males in each treatment group were mated, with two control females each, six weeks post-injection for three weeks. Males were euthanized immediately after the mating period (~10 weeks post-injection) **A.** Average testis weights (n = 10). **B.** Tubule areas. H and E stained sections were imaged and then at least 50 tubules from each of five males from each treatment group was circled and the area in pixels/tubule was calculated, using imageJ (n = 5, p<0.05). Statistical analysis by One-way ANOVA then post hoc using student T-tests. **C. Histology of the testes.** Representative images of haematoxylin and eosin stained sections illustrating relative area of tubules. Scale bar = 50 μm.

FSH2Menadione alone or the vehicle had testes weights over 126 mg. However, the overall mean weights of the testes of LH2Auristatin and FSH2Menadione/LH2Auristatin treated males were no different to that of the vehicle control group (Fig 4A).

To account for the lower testes weights, the morphology of fixed testis sections was examined. Qualitatively, the seminiferous tubules of the LH2Auristatin- and the FSH2Menadione/ LH2Auristatin-treated males appeared much smaller than those of the vehicle control or the FSH2Menadione-treated males (Fig 4B). Similarly, a quantitative evaluation using ImageJ software revealed that the average area occupied by seminiferous tubule in the testes of all of the treated animals were lower than that of the vehicle control group (Fig 4C). This reduction in tubule area proved highly significant ($P < 0.005$), such that the mice treated with LH2Auristatin and FSH2Menadione/LH2Auristatin exhibited a 40% reduction in mean tubule area compared to the testes of the control group. Such changes in tubule area were accompanied by the presence of apoptotic germ cells, which were clearly discerned in the two treatment groups (Fig 4B). To confirm germ cell apoptosis had indeed occurred, ApopTag was used to detect single-stranded and double-stranded breaks associated with this form of cell death. The results of this assay confirmed that the testes sections from both the LH2Auristatin and the FSH2Menadione/LH2Auristatin treatment groups exhibited significantly ($P < 0.05$) more tubules containing germ cells undergoing apoptosis (Fig 5A and 5B) and a significantly ($P < 0.05$) greater number of apoptotic cells per affected tubule (Fig 5C) than the FSH2Menadione alone or vehicle treatment groups Given the low number of apoptotic cells in the testes treated with FSH2Menadione alone, it appeared that the apoptosis was mostly precipitated by damage inflicted by the Auristatin.

## FSH2Menadione with LH2Auristatin had a small but not significant impact on Sertoli cell number

By targeting menadione to the testes using FSH2 peptide, we sought to ablate Sertoli cell function and vitality. To assess the success of this strategy, the number of Sertoli cells per seminiferous tubule were determined by immunolabeling of testes sections with antibodies directed against SOX9; a protein whose testicular expression is restricted to the Sertoli cell population. This analysis revealed a trend, albeit not statistically significant, of reduced mean numbers of Sertoli cells within the testes of the FSH2Menadione/LH2Auristatin treatment group (i.e. ~14 Sertoli cells/tubule) compared to the number of these cells in the vehicle control group (i.e. ~17 Sertoli cells/tubule) (S5B Fig). This trend was not replicated among Sertoli cell populations in the testes of males treated with either FSH2Menadione or LH2Auristatin, neither of which were numerically different from those treated with the vehicle. The lack of significant impact on the Sertoli cell population was reflected in the serum FSH levels, with an ELISA showing no difference in FSH levels within the cardiac sampled blood serum (S5C Fig) between the treatment groups. These data indicate that the targeted delivery of low dose menadione to Sertoli cells was not sufficient to create a significant reduction in their number.

## The mature spermatozoa of FSH2Menadione- and FSH2Menadione/ LH2Auristatin-treated males exhibited DNA strand breaks and oxidative DNA damage

Despite no overt effect on Sertoli cell populations, FSH2Menadione- and FSH2Menadione/ LH2Auristatin-treatment did elicit oxidative DNA damage among the mature spermatozoa of treated males. Indeed, the application of a HALO assay indicated that there was a significant increase ($P < 0.05$) in DNA strand breaks harboured by the spermatozoa from FSH2Menadione- and FSH2Menadione/LH2Auristatin-treated animals (Fig 6A) compared to that of the

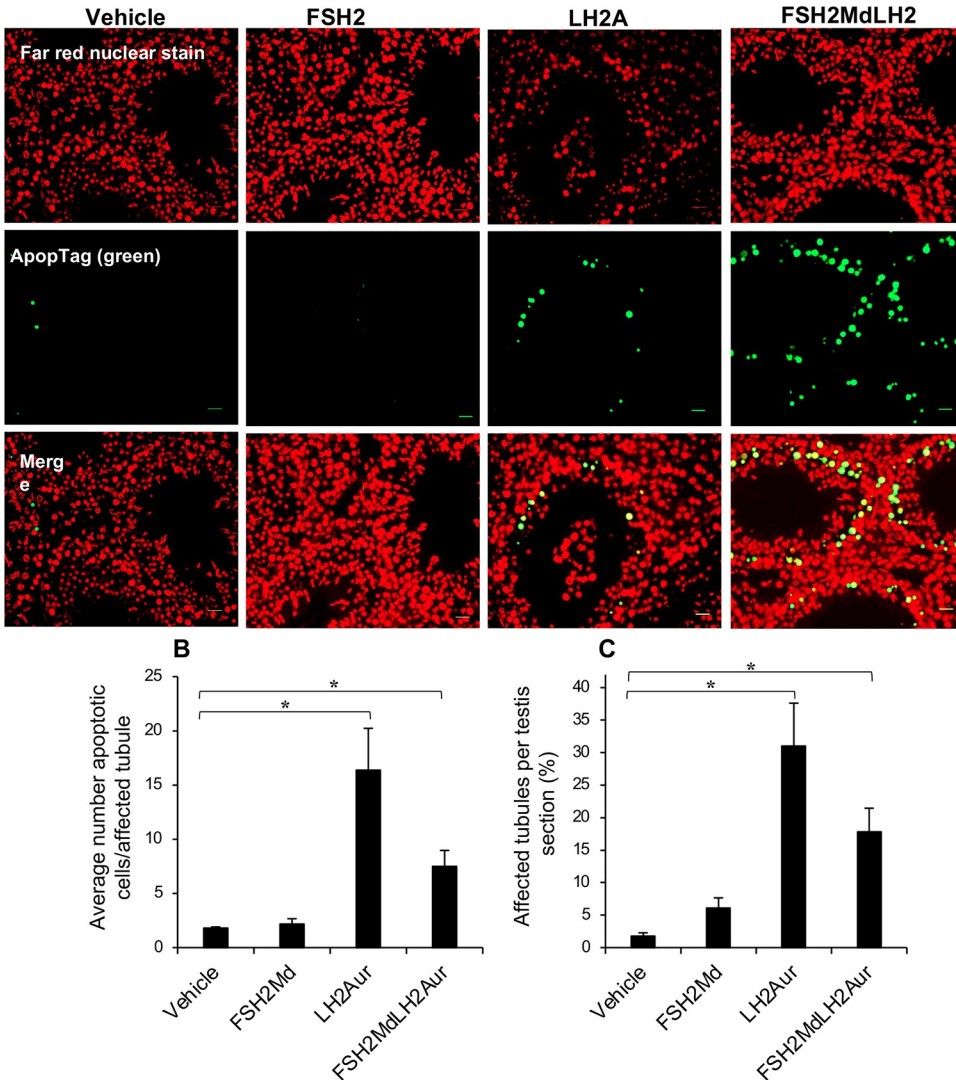

**Fig 5. Effect of FSH2Menadione (FSH2Md) and LH2Auristatin (LH2Aur) *in vivo*.** Male adult mice were injected IP with 300 µl/30 g of either 14.5 mM FSH2Menadione; 100 µl/30 g 420 µM LH2Auristatin; a combination of both, 16 hours apart; or 300 µl/30 g of the vehicle (30% Kolliphor/PBS) with 10 mice in each treatment group. Five males in each treatment group were mated, with two control females each, six weeks post-injection for three weeks. Males were euthanized immediately after the mating period (~10 weeks post-injection) **A.** Representative images of ApopTag. ApopTag staining was used to stain for cells undergoing apoptosis in sections from five males from each treatment group. Scale bar = 50 µm **B.** The total number of tubules/section (n = 5 sections/treatment group) and **C.** Percentage of tubules affected (>4 stained cells) were then counted along with the number of stained cells per affected tubule. NB. Subjects were randomly selected from the treatment groups but those testes weighing less than 60 mg were excluded. Statistical analysis by One-way ANOVA then post hoc using student T-tests.

control group. Specifically, an average of 22% of the spermatozoa from FSH2Menadione-treated and 17% from FSH2Menadione/LH2Auristatin-treated males displayed DNA strand breaks compared to only 5% of spermatozoa recovered from the cauda epididymis of LH2Auristatin- alone or vehicle-treated males. Additionally, there was a greater incidence of oxidation of DNA guanosine bases in the spermatozoa from the LH2Auristatin- and the FSH2Menadione/LH2Auristatin-treated treatment groups (averaged between 25 and 31%) compared to the vehicle-treated animals (Fig 6B). Furthermore, the oxidative DNA damage observed in the spermatozoa was evidently not a product of ongoing ROS production as there was no

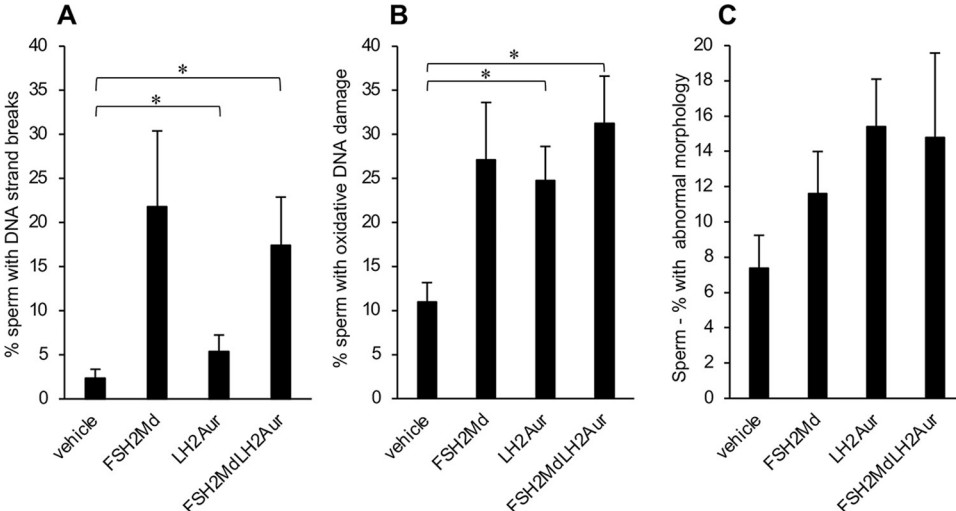

**Fig 6. Epididymal sperm–effect of FSH2Menadione (FSH2Md) and LH2Auristatin (LH2Aur)** *in vivo*. Male adult mice were injected IP with 300 µl/30 g of either 14.5 mM FSH2Menadione; 100 µl/30 g 420 µM LH2Auristatin; a combination of both, 16 hours apart; or 300 µl/30 g of the vehicle (30% Kolliphor/PBS) with 10 mice in each treatment group. Males were euthanized 10 weeks post-injection and epididymal sperm was collected from all ten males in each treatment group and all samples were assessed for abnormal morphology, vitality, motility, mitochondrial ROS, cytoplasmic ROS, and DNA fractionation with at least one hundred sperm counted in each assay for each sample. **A.** Sperm were probed for DNA strand breaks using the HALO assay and sperm were categorised as either having a halo, or no halo. **B.** An anti-8-hydroxy-deoxyguanosine antibody was used to detect oxidised DNA guanosine bases. **C.** Sperm smears of the sperm were fixed in methanol and then stained using the Rapid Diff Kit (Pathtech). Sperm were categorised as either having normal or abnormal morphology. Statistical analysis by One-way ANOVA then post hoc using student T-tests.

difference in the levels of either mitochondrial ROS or cytoplasmic superoxide detected within these cells (S6A and S6B Fig) between the four treatment groups. Moreover, the DNA damage burden in the caudal spermatozoa of treated males was not accompanied by a significant increase in the incidence of abnormal sperm morphology compared to the control males (Fig 6C). Computer-assisted sperm analysis (CASA) also showed no difference in sperm motility parameters or sperm total motility among the treated males compared to that of the vehicle controls (S7A–S7D Fig). Furthermore, there was no difference in the vitality of the caudal spermatozoa (S6C Fig) or in the DNA fragmentation index the sperm chromatin structure assay (SCSA) (S6D Fig) between the treatment groups. Notwithstanding these results, the DNA damage detected in the spermatozoa of some of the treatment groups appeared to be of sufficient magnitude to impinge on the fertility of the treated males, which was consequently assessed.

## The combination of LH2Auristatin and FSH2Menadione decreased the fertility of treated sires

Eighty percent of the females that had been housed with the LH2Auristatin or the FSH2Menadione/LH2Auristatin injected males became pregnant with live embryos present at 13 days gestation. A mating plug was observed in the remaining 20% of females but no embryos were present at 13 days post-mating. All ovaries exhibited corpora lutea, indicating that ovulation was likely to have taken place in these females. There were significantly fewer average embryos/pregnant female mated with males from LH2Auristatin and FSH2Menadione/LH2Auristatin treatment groups compared to those females mated with the vehicle control sires (Fig 7A). Additionally, 87.5% of females mated with males in the FSH2Menadione/

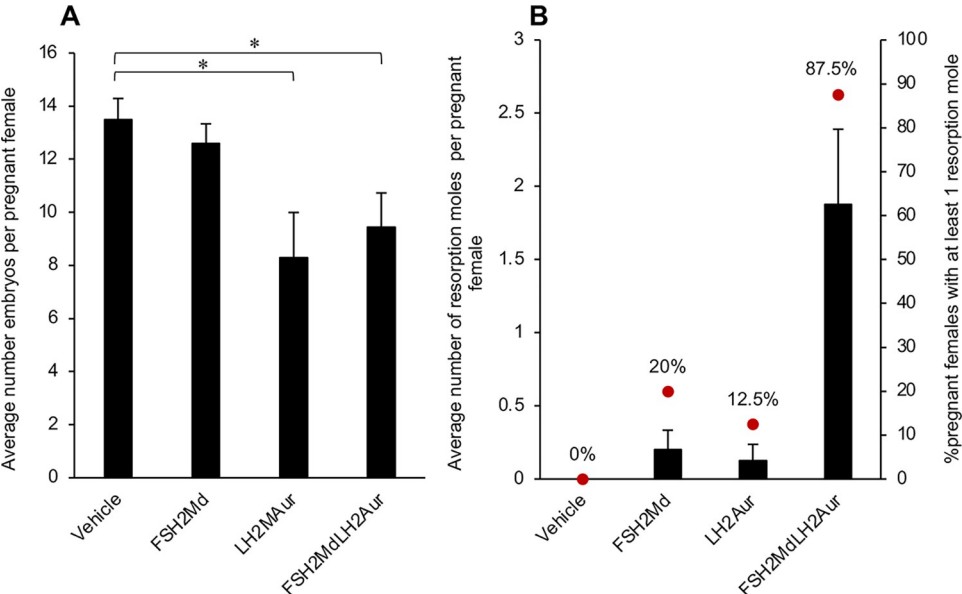

**Fig 7. Effect of FSH2Menadione (FSH2Md) and LH2Auristatin (LH2Aur) *in vivo* on fertility.** Male adult mice were injected IP with 300 μl/30 g of either 14.5 mM FSH2Menadione; 100 μl/30 g 420 μM LH2Auristatin; a combination of both, 16 hours apart; or 300 μl/30 g of the vehicle (30% Kolliphor/PBS) with 10 mice in each treatment group. Five males in each treatment group were mated, with two control females each, six weeks post-injection for three weeks. At 13 days post-coitus, females were euthanized then embryos and resorption moles were counted. **A.** The number of embryos per pregnant female. **B.** The average number of resorption moles per pregnant female and the proportion of pregnant females that had a least one resorption mole.

LH2Auristatin treatment group displayed evidence of embryo resorption, in the form of resorption moles, at 13 days post-mating (Fig 7B). Whilst all treatment groups, produced a higher than average number of resorption moles than that of the vehicle control group, the FSH2Menadione/LH2Auristatin treatment in particular produced a very high, 1.9 resorption moles/pregnant female (Fig 7B), with one female mated from this group exhibiting five resorption moles.

FSH2Menadione administration alone was not sufficient to compromise male fertility. In fact, all five males in the FSH2Menadione treatment group were fertile, with each of the ten females housed with these males becoming pregnant and carrying live embryos at 13 days post-mating (Fig 7A). Two of the females that had been mated with males from this treatment group exhibited one resorption mole each, with none detected in the remaining eight dams. In contrast to the above, all five males in the vehicle treatment group proved to be fertile with those females that became pregnant having at least 11 (up to 17) and an average of 13.5 embryos present at 13 days, none had resorption moles (Fig 7A).

## Discussion

There is an increasing demand for humane, accessible, and cost-effective non-surgical sterilization methods for the control of populations of both domestic and free-roaming cats and dogs as well as feral populations of mammalian pest animal species. An injectable sterilization agent would not only act as a boon for animal rescue organizations but also for those countries and remote areas where access to surgical sterilization is restricted. In addressing this need, there is a clear imperative to develop an agent that will provide targeted and permanent ablation of those cells involved in gametogenesis. A targeted agent should also offer species specificity in the absence of adverse off-target effects. Here, we have designed homing peptides

capable of selectively delivering cell ablation payloads to the niche environment in which the male germ line develops. Specifically, we have engaged in the rational design of LHr-targeting peptides to complement a previously developed FSHr-targeting peptide. The FSHr-targeting peptide (FSH2) was based on the receptor's crystal structure [37] and exploited prior knowledge regarding the FSH amino acids implicated as FSHr ligands [24, 37, 42]. The LHr-targeting peptides were developed in a similar fashion with the LHa peptide proving to be the most suitable for our purposes.

From a general perspective, one of the most significant outcomes of this *in vivo* study was that ten weeks, or two complete spermatogenic cycles [43], after a single administration of FSH2Menadione and LH2Auristatin, the mature epididymal sperm of treated males exhibited a significantly higher level of DNA strand breaks and oxidative DNA damage than that of their control counterparts. Although spermatogenesis and fertility persisted in some animals, the treatments facilitated enduring damage to the testes and confirmed that the Sertoli cell and Leydig cell populations had been targeted. At these concentrations and with a single dose of each, LH2Aur had worked synergistically with FSH2Md to impact testicular function and, ultimately, decrease the viability of the embryos. However, the sterilizing impact of these constructs lacked consistency. So, while this combination of peptide-based reagents does not yet constitute the perfect sterilization package, the results demonstrate the general feasibility of this approach and suggest areas where more work needs to be done.

Ideally, to minimize the possibility of adverse side effects, a targeting peptide should minimize accumulation in off-target tissues [44]. Of the four LH peptide-FITC conjugates the peptide with the highest binding capability *in vitro*, L95, along with L101 and LHa were designed to encompass the sequence of the "seatbelt" region of LH β-subunit. L57 peptide, representative of residues residing within the βL2 loop binding region of the β-subunit of LH (Table 1, Fig 1), displayed higher binding capability than either L101-FITC or LHa-FITC *in vitro*. Since both the "seatbelt" and the βL2 loop regions of LHβ have been implicated in ligand binding to the cognate LH receptor [23, 42], it is considered likely that differences in *in vitro* peptide binding efficacy are attributed to their unique physicochemical properties. L95 peptide, despite displaying a higher binding capability, also lost specificity at higher concentrations (Fig 2B). This increase in binding capability and lack of specificity at higher concentrations is most likely attributable to the relatively high positive charge of +2.9 on this peptide at neutral pH (Table 1). A higher positive charge would enable this peptide to more readily bind non-specifically via electrostatic attraction to the negatively charged cell membrane in this *in vitro* system [45]. Additionally, L57 with a charge of +1.9, exhibited a higher relative binding capability then the LHa- and L101-FITC conjugates that respectively had predicted charges of +1 and +0.9 (Table 1, Fig 2B). Ultimately, such charge differences did not translate directly to an *in vivo* setting, in which all four of the peptides displayed a similar localization to the testes.

Given its performance *in vivo*, we chose to move forward with the LHa peptide expressing an N-terminal Linker 2 (hereinafter referred to as LH2), to deliver our chosen anticancer cell ablation agent, Auristatin (S4 Fig) to the target cell population. For this study, we utilized a hydrophobic form of Auristatin that readily enters the cell and, once within the cell, acts as an antimitotic agent, binding to α-tubulin and preventing the formation of a mitotic spindle, leading to G2-M cell cycle arrest. Auristatin is highly potent and so, it is important that this drug is delivered selectively by conjugation to a targeting agent (most commonly an antibody) so as to reduce adverse off-target effects. Therefore, Auristatin was attached to the LHa peptide via a cathepsin cleavable valine-citrulline(vc)-PABC linker (S4 Fig). This linker has been reported to have high stability in serum or plasma and, once the peptide-Auristatin binds to the LH receptor, it should be taken into the cell where the complex will be metabolised by lysosomal proteases [46] followed by 1,6-elimination to liberate the active drug (S4 Fig).

Our aim was to target both the Leydig and the Sertoli cells concurrently and, so, along with the LH2Auristatin, to target the Leydig cells, we used a FSH2 peptide carrying 2-methyl-1,4-naphthoquinone (menadione) to attack the Sertoli cells. On entering the cell, the Auristatin is freed from the peptide and can be released from the cell. Free Auristatin can then kill surrounding cells via the bystander effect [47, 48]. For its part, the FSH2Menadione reagent was designed to enter Sertoli cells and create a localized oxidative stress that would destroy the stem cell niche as well as all stages of spermatogenesis [49, 50]. Spermatogonial germ cells are exquisitely sensitive to the effects of ionizing radiation [51] that induces DNA strand breaks mainly via the radiolysis of water and the resulting formation of hydroxyl radicals [52]. Ionizing radiation also generates other reactive oxygen species (ROS), such as superoxide and hydrogen peroxide, all of which can interact with DNA, cellular lipids and proteins [53] thus facilitating irreversible damage to the stem cell niche. Addition of the redox-cycling reagent, menadione, was an attempt to recapitulate these radiation effects through the enzymatic generation of superoxide anion and hydrogen peroxide [54]. Hydrogen peroxide is converted through a series of reactions to hydroxyl and hydroperoxyl radicals that, in turn, act as oxidizing agents. Together with other reactive oxygen species (ROS), the radicals so formed can cause loss of function through oxidative damage to cellular lipids [55], proteins, and DNA [56]. Indeed, a previous study found that the application of FSH2Menadione led to oxidative DNA damage and DNA strand breaks in the spermatozoa of treated mice [22]. The presence of ROS can also induce oxidative actin cross-linking and dissociation of the cytoskeleton from the plasma membrane. All these alterations appear to contribute to the multifactorial process underlying the irreversible cell injury caused by oxidative stress. Additionally, as well as its antimitotic effects, Auristatin is a known radiosensitizing agent that renders affected cells more vulnerable to oxidative stress [57–59]. The strategy for this study was therefore to challenge both the Sertoli cell and Leydig populations concurrently by the production of localised oxidative stress in the vicinity of the germ cells, in the presence of an anti-mitotic agent that may deplete the Leydig cell population as well as enhance the impact of ROS within the testes.

In accounting for the results observed in LH2Auristatin treated males, Auristatin is a very potent tubulin inhibitor. It binds to microtubules, introduces structural defects, and suppresses microtubule (MT) dynamics, leading to the suppression of proliferation, mitosis, and a disrupted MT network [60]. A depleted Leydig cell population means that their support of Sertoli cell health is disrupted. Additionally, the cytoskeleton is responsible for movement of organelles and molecules involved in steroidogenesis (17-β-estradiol synthesis) [61] in the Leydig cells. If, as a result, local testosterone levels are reduced then the ability of Sertoli cells to support spermatogenesis [62] might be disrupted, contributing to the apoptosis of the germ cells. We did not measure testosterone levels in this particular study, however any further exploration of this approach to sterilization should certainly include such measurements. If the Auristatin released from the Leydig cells also reaches the Sertoli cell population, then the resulting suppression of MT dynamics, might also have a direct disruptive action on the germ cell niche, adversely affecting the entire germ cell population [63, 64]. Indeed, all stages of spermatogenesis are reliant on the dynamic organization/reorganization of microtubules [65] and the cytoskeleton [66]. Spermatozoa from LH2Auristatin-treated males also displayed a higher level of oxidative damage than the controls, similar to that of the FSH2Menadione treated males. This might have been dependent on MT disruption since 8-Oxoguanine DNA Glycosylase (OGG1), which excises the oxidized bases from DNA, associates with and relies on microtubules for transport to the site of action during mitosis [67].

The high number of resorption moles in females housed with FSH2Menadione and LH2Aur-treated sires (Fig 7B) is most likely a consequence of DNA damage to the spermatozoa precipitated by the presence of ROS, produced by the menadione and further potentiated

by the disruptive action of Auristatin. However, we can exclude other male-mediated effects, possibly secondary to oxidative stress, including epigenetic impacts on non-coding RNA species carried into the oocyte by the fertilizing spermatozoon or the disrupted ability of sperm centrioles to orchestrate cell division in the offspring. In order to better understand the etiology of embryonic loss in this model detailed studies would have to conducted at early stages of development. At this stage, we can only state that our observations are consistent with the known involvement of sperm DNA damage a in the reduction in litter size and increased— number of embryonic resorptions/dam induced induced by cytotoxic reagents *in vivo* [68]. Similarly, transient testicular heating results in increased oxidative DNA damage in the germ line which again precipitates an increased incidence of embryonic resorption sites following mating [69, 70].

In accounting for the variation in response within treatment groups, there are several factors that may adversely impact the amount of Auristatin or menadione that is delivered to the target cells. Peptides can have a short *in vivo* half-life due to rapid clearance by the kidneys and degradation by endogenous proteases [40]. This can severely limit the proportion of peptide conjugate that reaches the target cells. One method of increasing *in vivo* stability is to increase the hydrodynamic radius of the peptide by the addition of polyethylene glycol [40] or fusion with albumin [41, 71] or other serum proteins [72]. However, this, along with other modifications to the peptide such as replacement of labile amino acids or replacement of L-amino acids with D-amino acids [40], may disrupt the binding capabilities of the peptide to its cognate receptor. Additionally, the vc-PABC linker that attaches the Auristatin to the peptide must be stable *in vivo*. Extracellular cleavage of the vc-PABC linker can be catalysed by carboxylesterase 1C in mouse serum, thereby releasing the Auristatin before it reaches the target cell [73]. Additionally, the toxicity of Auristatin necessitated the delivery of a very low final concentration of 1.42 μM, which may have impacted the efficacy of our approach *in vivo*. Interindividual variation in the responses to menadione might also reflect small variations between animals in the availability of transition metals or the isoform profile of ROS-metabolizing enzymes in this outbred mouse strain [74].

## Conclusions

In summary, we have successfully designed an LHr-targeting peptide and demonstrated selective binding to a Leydig cell line *in vitro* as well as specific targeting *in vivo*. We have applied this LHr-targeting peptide to deliver Auristatin to the Leydig cells *in vivo*, both alone and concurrently with a FSHr-targeting peptide delivering menadione to the Sertoli cells. The results were promising in terms of precipitating germ cell apoptosis but the outcomes within the treatment groups were varied. The variation in responsiveness may have stemmed from enzymatic degradation of the peptide and/or the vc-PABC linker *in vivo*, resulting in a suboptimal quantity of Auristatin reaching the target cell. Whatever the cause, the higher level of embryo resorptions exhibited in females housed with LH2Auristatin/FSH2Menadione treated sires would not be acceptable clinically. Thus while, in its present state, this technology cannot compete with hormonal methods that we know are efficacious and well tolerated, the results obtained in this study do support the future use of peptides as vectors to remove non-renewable cell types from the reproductive system and effect complete sterilization with a single administration. Recently, untargeted nanoparticles have been used to encapsulate cytotoxic agents and have been found to be efficacious for testis cell ablation in *in vitro* and *ex vivo* settings [75, 76]. Attachment of the targeting peptides to the surface of such nanosystems, offers the advantage of simultaneously protecting the peptide from clearance and degradation and of increasing the amount of cargo that can be delivered. This targeted approach has the potential

to increase local concentrations of the reagents as well as avoiding potential off-target effects. Given the promising results of this study, delivery of menadione and Auristatin within targeted nanoparticles as a single injection may increase the local concentration of these two drugs to a level that will elicit a more pronounced and consistent effect.

## Supporting information

**S1 Fig. Quantitative PCR (qPCR) primers.**
(DOCX)

**S2 Fig. Choice of Leydig cell model. A.** Relative expression of LHr mRNA in mouse Leydig cell lines, MLTC1 and TM3 cells, whole testis and mEcap18 cells assessed by qPCR. **B.** Immunocytochemistry using an anti-LHr antibody with an AlexFluor 488 secondary indicates that the LH receptor is expressed at high levels by MLTC1 cells but negligibly in mEcap18 cells.
(DOCX)

**S3 Fig. Representative images of excretory organs (kidney, liver and spleen) 8 hours post-injection.** Male adult mice were injected IP with 300 μl of either 1mM peptide-FITC in 30% Kolliphor/PBS or 30% Kolliphor/PBS with 3 mice in each treatment group. Animals were euthanized 8 hours post-injection and all tissues were imaged using Leica MZFLIII microscope with an epifluorescent green filter next to the vehicle control. Peptide-FITC applied is indicated in the first image of each row. Fourth tissue in the bottom three images were an unrelated peptide-FITC.
(DOCX)

**S4 Fig. Lysosomal processing of vcMonomethylAuristatin E (vcMMAE), an anticancer agent that interacts with α-tubulin in a similar way to Vinca alkaloids to block α-tubulin polymerisation and prevents the formation of the mitotic apparatus.** The linked maleimide allows conjugation of the peptide via the N-terminal thiol group. Between the maleimide and the Auristatin there is a cleavable linker made up of valine, citrulline (circled in red) and para-aminobenzyl carbamate (PABC). In the lysosome, the dipeptide linker is cleaved and then Auristatin is released from PABC by 1,6-elimination. Structures drawn using software: Advanced Chemistry Development Inc. ACD/3D Viewer (Freeware) Product Version 12.01 (Build 32890, 18 May 2009).
(DOCX)

**S5 Fig. Effect of FSH2Md and LH2Auristatin *in vivo*.** Male adult mice were injected IP with 300 μl/30 g of either 14.5 mM FSH2Menadione; 100 μl/30 g 420 μM LH2Auristatin; a combination of both, 16 hours apart; or 300 μl/30 g of the vehicle (30% Kolliphor/PBS) with 10 mice in each treatment group. Five males in each treatment group were mated, with two control females each, six weeks post-injection for three weeks. Males were euthanized immediately after the mating period (~10 weeks post-injection) **A.** Anti-Sox9 was used to stain for Sertoli cells in testis sections from each of five males in each treatment group. Images of the testes of the treated males. Inset is secondary only control. Scale bar = 50 μm. **B.** Sertoli cells were counted in all tubules of each section for 5 individuals from each treatment (n = 5). NB. Subjects were randomly selected from the treatment groups but excluded those testes weighing less than 60 mg. **C.** Serum FSH levels were measured using an anti-FSH ELISA (n = 9).
(DOCX)

**S6 Fig. Epididymal sperm–effect of FSH2Menadione (FSH2Md) and LH2Auristatin (LH2Aur) *in vivo*.** Male adult mice were injected IP with 300 μl/30 g of either 14.5 mM FSH2Menadione; 100 μl/30 g 420 μM LH2Auristatin; a combination of both, 16 hours apart;

or 300 μl/30 g of the vehicle (30% Kolliphor/PBS) with 10 mice in each treatment group. Males were euthanized 10 weeks post-injection and epididymal sperm was collected from all ten males in each treatment group and all samples were assessed for vitality, motility, mitochondrial ROS, and cytoplasmic ROS. Flow cytometric analysis was used with probes to assess A. Mitochondrial ROS (MSR) and **B.** Cytoplasmic superoxide (DHE). **C.** The eosin exclusion method was used to assess the caudal sperm vitality. **D.** The sperm chromatin structure assay (SCSA) detected susceptibility to acid-mediated DNA fragmentation expressed here as DNA fragmentation index (DFI). Statistical analysis by One-way ANOVA showed no significant differences between the treatments.
(DOCX)

**S7 Fig. Epididymal sperm–effect of FSH2Menadione (FSH2Md) and LH2Auristatin (LH2Aur)** *in vivo*. Male adult mice were injected IP with 300 μl/30 g of either 14.5 mM FSH2Menadione; 100 μl/30 g 420 μM LH2Auristatin; a combination of both, 16 hours apart; or 300 μl/30 g of the vehicle (30% Kolliphor/PBS) with 10 mice in each treatment group. Males were euthanized 10 weeks post-injection and epididymal sperm was collected from all ten males in each treatment group and the motility all samples were assessed using CASA. **A.** Total motility. **B.** Motility**. C.** progressive motility. **D.** Slow motility. Statistical analysis by One-way ANOVA showed no significant differences between the treatments.
(DOCX)

**S8 Fig. Effect of FSH2Menadione (FSH2Md) and LH2Auristatin (LH2Aur)** *in vivo*. Male adult mice were injected IP with 300 μl/30 g of either 14.5 mM FSH2Menadione; 100 μl/30 g 420 μM LH2Auristatin; a combination of both, 16 hours apart; or 300 μl/30 g of the vehicle (30% Kolliphor/PBS) with 10 mice in each treatment group. Five males in each treatment group were mated, with two control females each, six weeks post-injection for three weeks. Males were euthanized immediately after the mating period (~10 weeks post-injection). Haematoxylin and eosin stained sections illustrating**. A.** Normal development of the seminiferous tubules in a vehicle control testis section and **B.** Abnormal development of the seminiferous epithelia in a testis section from FSH2MdLH2Aur-treated male. Scale bar = 100 μm.
(DOCX)

## Acknowledgments

The authors thank Natalie Regan, Annalucia Darbey, Anne-Louise Gannon, Lee Smith, Natalie Trigg, and Kasey King for technical assistance; Nathan Smith for conducting liquid chromatography-mass spectrometry; Dr Petra Sipilä for gift of mECap18 cells.

## Author Contributions

**Conceptualization:** Brett Nixon, Robert John Aitken.

**Data curation:** Barbara Fraser.

**Formal analysis:** Barbara Fraser.

**Funding acquisition:** Robert John Aitken.

**Investigation:** Alex Wilkins, Sara Whiting, Mingtao Liang, Diane Rebourcet, Brett Nixon.

**Methodology:** Alex Wilkins, Sara Whiting, Mingtao Liang, Diane Rebourcet.

**Project administration:** Robert John Aitken.

**Supervision:** Brett Nixon.

**Writing – original draft:** Barbara Fraser.

**Writing – review & editing:** Brett Nixon, Robert John Aitken.

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
