## [Decision Letter · Decision Letter 0]

17 Oct 2023

PONE-D-23-29546Development of peptides for targeting cell ablation agents concurrently to the Sertoli and Leydig cell populations of the testes: an approach to non-surgical sterilizationPLOS ONE

Dear Dr. Aitken,

Thank you for submitting your manuscript to PLOS ONE. After careful consideration, we feel that it has merit but does not fully meet PLOS ONE’s publication criteria as it currently stands. Therefore, we invite you to submit a revised version of the manuscript that addresses the points raised during the review process.

The study is quite interesting, especially in the methods to target specific types of cells in the testis. There are many concerns, especially the rationale as to how this methodology is superior to the already existing ones. Further, detailed methodology should be provided for better understanding of the experimental procedures. Bibliography should be up to date with all details.

We look forward to receiving your revised manuscript.

Kind regards,

Suresh Yenugu

Academic Editor

PLOS ONE

Journal Requirements:

Reviewers' comments:

Reviewer's Responses to Questions

**Comments to the Author**

1. Is the manuscript technically sound, and do the data support the conclusions?

Reviewer #1: Yes

Reviewer #2: Partly

2. Has the statistical analysis been performed appropriately and rigorously? 

Reviewer #1: Yes

Reviewer #2: Yes

3. Have the authors made all data underlying the findings in their manuscript fully available?

Reviewer #1: Yes

Reviewer #2: No

4. Is the manuscript presented in an intelligible fashion and written in standard English?

Reviewer #1: No

Reviewer #2: Yes

5. Review Comments to the Author

Reviewer #1: Development of peptides for targeting cell insertion agents coincidentally to the Sertoli and Leydig cell populations of the tests: an approach to non-surgical sterilization, this is a valuable study. The design of this article is reasonable. There are only some minor issues, and the language needs further modification. The pictures in the text need to be replaced, especially Figure 5a. Figure 3a needs to be replaced.

Reviewer #2: The manuscript „Development of peptides for targeting cell ablation agents concurrently to the Sertoli and Leydig cell populations of the testes: an approach to non-surgical sterilization” describes pre-experiments for the development of a new non-surgical sterilization method for dogs and cats. Stray animals are a growing problem in many countries and existing hormonal sterilizations methods as well as conventional surgery are not always available/feasible in all circumstances. The authors created and used recombinant proteins targeting the LH receptor present on Leydig cells (either cell culture or in vivo in mice) to administer Auristatin, a anti-cancer drug, to Leydig cells. By disturbing/disrupting Leydig cell function, germ cell apoptosis was detected. It was made clear, that LH2Auristatin alone had less effect than a combinantion of LH2Auristatin/FSH2Menadione. The manuscript is well-written and experiments performed to develop LH2Auristatin are sound. Nevertheless, there is a main issue in my opinion: The experiments on FSH2Menadione rely on ref. 24, which is described as “to be submitted, 2021” in the reference list. I tried to find the paper on Pubmed (in case the reference software was making a mistake) but didn’t find it. So, at least half of the manuscript is based on a not publically available manuscript which is not suitable. Please comment/revise this.

Moreover, I would like the authors to comment on the advantages of their approach over existing hormonal contraceptives such as GnRH implants; the authors discuss that this method requires an injection and is not “long-lasting”. But their LH2Auristatin/FSH2Menadione approach also requires two injections (both peptides 16 hours apart) and there a no data on long-time effects. Also, the peptides are administerd by IP injection – what would be the route of administration in dogs and cats? Additionally, the data shown on germ cell apoptosis are not sufficient, the pictures are too small to see which germ cells undergo apoptosis. As there ARE sperm produced and pups generated, the methods doesn’t seem efficient enough. Embryos are generated, and embryonic resorption happens. Are the treated sires completely sterile or are also pups born? If so, your methods stands behind GnRH analoga.

There are also other (partly minor) issues that have to be addressed. Detailed comments can be taken from the attached PDF. Here are some points to take care about:

1) References: Ref. 4 seems to be incorrect. Please check and revise. As mentioned above, Ref. 24 cites a not publically available manuscript that has (or has not?) been submitted but not published. Please also check the reference list for formatting, sometimes brackets are there but not closed. Two references in the text are cited with “name, YOP” (Sipilä et al., 2004 and Hendrickson, 2005). For sperm analysis, please cite the recent WHO manual.

2) qPCR: Please provide more information on a) primers, b) qPCR procedure, and c) evaluation of data. Follow MIQUE guidelines (Bustin et al. 2009) for reference. Missing information is e.g. Ref Seq, amplicon length, annealing temperature, melt curve data, primer efficiency etc.

3) IHC, page 10: Please add more information on antibody dilution, host species etc. How thick were the sections used for IHC?

4) Figures: Please add reference “Fig. 4A” to line 544. Did I miss reference or figures for other organs’ weights (ines 544ff)? Fig. 4B is too small to see apoptotic germ cells; it would be interesting to know, WHICH germ cells are affected. Is it only spermatogonia or also developed germ cells? Also S5 Fig. B is very small; stained Sertoli cells can be imagined, but not really seen. Did the authors also check for Leydig cell numbers, e.g. using CYP11A1 as marker? You cite S7 Fig. before S6 Fig., please revise.

5) Leydig cell function: The authors state, that disturbed Leydig cell function is responsible for germ cell disruption without adverse effects; have they checked for systemic testosterone levels and their impact on muscle mass etc?

6) ROS/DNA damage: The authors conclude that embryonic loss is due to DNA damage in sperm; I don't understand and agree with this conclusion. Embryonic loss means that there were sperm present to fertilize the egg. A later disruption of embryonic development not neccessarily is connected to germ cell disruption; have the authors checked the developing embryos for any cellular disorganization/malformations?

6. PLOS authors have the option to publish the peer review history of their article (what does this mean?). If published, this will include your full peer review and any attached files.

Reviewer #1: No

Reviewer #2: No

---

## [Author Response · Author response to Decision Letter 0]

8 Dec 2023

Reviewers' comments:

Reviewer's Responses to Questions

Comments to the Author

1. Is the manuscript technically sound, and do the data support the conclusions?

Reviewer #1: Yes

Reviewer #2: Partly

2. Has the statistical analysis been performed appropriately and rigorously? 

Reviewer #1: Yes

Reviewer #2: Yes

3. Have the authors made all data underlying the findings in their manuscript fully available?

Reviewer #1: Yes

Reviewer #2: No

4. Is the manuscript presented in an intelligible fashion and written in standard English?

Reviewer #1: No

Reviewer #2: Yes

5. Review Comments to the Author

Reviewer #1: Development of peptides for targeting cell insertion agents coincidentally to the Sertoli and Leydig cell populations of the tests: an approach to non-surgical sterilization, this is a valuable study. The design of this article is reasonable. There are only some minor issues, and the language needs further modification. 

Response: The revised manuscript has been carefully edited- so hopefully there are no remaining problems with the language. 

 Pictures in the text need to be replaced, especially Figure 5a. Figure 3a needs to be replaced.

Response: Figures 5 and 3 has been revised accordingly.

Reviewer #2: The manuscript „Development of peptides for targeting cell ablation agents concurrently to the Sertoli and Leydig cell populations of the testes: an approach to non-surgical sterilization” describes pre-experiments for the development of a new non-surgical sterilization method for dogs and cats. Stray animals are a growing problem in many countries and existing hormonal sterilizations methods as well as conventional surgery are not always available/feasible in all circumstances. The authors created and used recombinant proteins targeting the LH receptor present on Leydig cells (either cell culture or in vivo in mice) to administer Auristatin, a anti-cancer drug, to Leydig cells. By disturbing/disrupting Leydig cell function, germ cell apoptosis was detected. It was made clear, that LH2Auristatin alone had less effect than a combination of LH2Auristatin/FSH2Menadione. The manuscript is well-written and experiments performed to develop LH2Auristatin are sound. Nevertheless, there is a main issue in my opinion: The experiments on FSH2Menadione rely on ref. 24, which is described as “to be submitted, 2021” in the reference list. I tried to find the paper on Pubmed (in case the reference software was making a mistake) but didn’t find it. So, at least half of the manuscript is based on a not publically available manuscript which is not suitable. Please comment/revise this. 

Response: We apologise for this error and are sorry that this has given you extra work. The data for FSH2Menadione has been published (see reference 22); we had updated this reference in some but not all of the text.

Moreover, I would like the authors to comment on the advantages of their approach over existing hormonal contraceptives such as GnRH implants; the authors discuss that this method requires an injection and is not “long-lasting”. But their LH2Auristatin/FSH2Menadione approach also requires two injections (both peptides 16 hours apart) and there a no data on long-time effects. Also, the peptides are administered by IP injection – what would be the route of administration in dogs and cats? 

Response: We administered two injections 16 hours apart because the two compounds could only be solubilised in water by the addition of 30% Kolliphor, which may cause diarrhoea in the mice at higher doses. As we wished to minimise any confounding side effects, we staggered the administration of the compounds. We anticipate that, if these compounds were to be further developed, that they would be reformulated in such a way to allow concurrent administration – most likely as part of a nanosystem with the active components encapsulated within a nanoparticle that would have the targeting peptides attached to their external surface. This type of formulation is also anticipated to have a longer in vivo half-life increasing the likelihood of greater penetration of the active reagents and increasing the consistency of the results. 

Line 251: Reviewer #2 asked if further time points (greater than 8 hours) were planned for looking at peptide-FITC localisation in vivo. We selected the 8 hour time point after preliminary experiments found that maximum localisation occurred at 8 hours post-injection and that the FITC fluorescence was almost fully dissipated at 16 hours post-injection. 

Additionally, the data shown on germ cell apoptosis are not sufficient, the pictures are too small to see which germ cells undergo apoptosis. 

Response: Figure 5 has now been revised to better illustrate the apoptotic germ cells.

As there ARE sperm produced and pups generated, the methods doesn’t seem efficient enough. Embryos are generated, and embryonic resorption happens. Are the treated sires completely sterile or are also pups born? If so, your methods stands behind GnRH analoga.

Response: This is an excellent point which we have addressed in our Conclusions [line 855]. The referee is absolutely correct in saying that our technology cannot compete with hormonal methods that have a tried and trusted track record. Nevertheless, the results we have generated suggest that there is some potential for the future use of LH and FSH peptides as vectors in achieving non-surgical sterilization – even if we are not there yet.

There are also other (partly minor) issues that have to be addressed. Detailed comments can be taken from the attached PDF. 

Response: In line 64: the keyword in this sentence is “relatively” – to further clarify we have altered Line 64 to read: “However, relative to an injection, surgical sterilization is labor intensive and expensive”

Here are some points to take care about:

1) References: Ref. 4 seems to be incorrect. Please check and revise. 

Response: Ref. 4 has now been changed to the correct reference.

 As mentioned above, Ref. 24 cites a not publically available manuscript that has (or has not?) been submitted but not published. Please also check the reference list for formatting, sometimes brackets are there but not closed. Two references in the text are cited with “name, YOP” (Sipilä et al., 2004 and Hendrickson, 2005).

Response: These references have now been amended.

For sperm analysis, please cite the recent WHO manual.

Response: This citation has now been changed (Ref. 32) to the 6th Edition 2021.

2) qPCR: Please provide more information on a) primers, b) qPCR procedure, and c) evaluation of data. Follow MIQUE guidelines (Bustin et al. 2009) for reference. Missing information is e.g. Ref Seq, amplicon length, annealing temperature, melt curve data, primer efficiency etc.

Response: Additional information has been provided in the revised Materials and Methods section. 

3) IHC, page 10: Please add more information on antibody dilution, host species etc.

Response: This has now been added Line 232 et seq

 How thick were the sections used for IHC?

Response: Line 326 has been amended to include section thickness of 5 µm.

4) Figures: Please add reference “Fig. 4A” to line 544. Did I miss reference or figures for other organs’ weights (ines 544ff)? 

Response: Apologies, it was not appropriate to cite Fig 4A in this position as Figure 4A represents the mean weights of the testes. However,“Fig 4A” has now been added in Line # 55 . As there was no difference observed in the body weights or the tissues (other than the testes) between the treated males and controls, these data are not included as a separate figure. However, the raw data files will be made publicly available on acceptance of the article for publication. 

Reviewer #2 also asked if there was a figure for the LC-MS result (Line #38). These data will also be included with the raw data files. 

Fig. 4B is too small to see apoptotic germ cells; it would be interesting to know, WHICH germ cells are affected.Is it only spermatogonia or also developed germ cells? Also S5 Fig. B is very small; stained Sertoli cells can be imagined, but not really seen. 

Response: Figures 4B and S5 Fig. B have now been revised accordingly.

Did the authors also check for Leydig cell numbers, e.g. using CYP11A1 as marker?

Response: In adult testes Leydig Cells (LC) are in a quiescent status but have the ability to regenerate from progenitors in case of depletion (PMID: 27743991 and PMID: 26446427). That is why targeting the LC alone would have been insufficient to induce permanent sterility, and the approach we developed involved the targeting of both LCs and Sertoli cells (SC). 

You cite S7 Fig. before S6 Fig., please revise.

Response: We have now revised these figure citations (Lines 552-560) and the order of the figures within the Supplementary data file.

5) Leydig cell function: The authors state, that disturbed Leydig cell function is responsible for germ cell disruption without adverse effects; have they checked for systemic testosterone levels and their impact on muscle mass etc? 

Response: The impact of LH-auristatin on testosterone levels was not undertaken in this study. However, in the revised Discussion we emphasize the importance of conducting such assessments in the future (line 800). 

6) ROS/DNA damage: The authors conclude that embryonic loss is due to DNA damage in sperm; I don't understand and agree with this conclusion. Embryonic loss means that there were sperm present to fertilize the egg. A later disruption of embryonic development not necessarily is connected to germ cell disruption; have the authors checked the developing embryos for any cellular disorganization/malformations?

Response: This interesting point has been addressed in the Discussion, line 815 et seq. There may have been many reasons why a damaged spermatozoon cell might precipitate embryonic loss as well as DNA fragmentation – for example, damage to non-coding RNA species or a loss of normal centriolar function. 

6. PLOS authors have the option to publish the peer review history of their article (what does this mean?). If published, this will include your full peer review and any attached files.

Do you want your identity to be public for this peer review? For information about this choice, including consent withdrawal, please see our Privacy Policy.

Reviewer #1: No

Reviewer #2: No

In compliance with data protection regulations, you may request that we remove your personal registration details at any time. (Remove my information/details)

---

## [Decision Letter · Decision Letter 1]

28 Dec 2023

Development of peptides for targeting cell ablation agents concurrently to the Sertoli and Leydig cell populations of the testes: an approach to non-surgical sterilization

PONE-D-23-29546R1

Dear Dr. Aitken,

We’re pleased to inform you that your manuscript has been judged scientifically suitable for publication and will be formally accepted for publication once it meets all outstanding technical requirements.

Kind regards,

Suresh Yenugu

Academic Editor

PLOS ONE

Additional Editor Comments (optional):

Reviewers' comments:

Reviewer's Responses to Questions

**Comments to the Author**

1. If the authors have adequately addressed your comments raised in a previous round of review and you feel that this manuscript is now acceptable for publication, you may indicate that here to bypass the “Comments to the Author” section, enter your conflict of interest statement in the “Confidential to Editor” section, and submit your "Accept" recommendation.

Reviewer #2: All comments have been addressed

2. Is the manuscript technically sound, and do the data support the conclusions?

Reviewer #2: (No Response)

3. Has the statistical analysis been performed appropriately and rigorously? 

Reviewer #2: (No Response)

4. Have the authors made all data underlying the findings in their manuscript fully available?

Reviewer #2: (No Response)

5. Is the manuscript presented in an intelligible fashion and written in standard English?

Reviewer #2: (No Response)

6. Review Comments to the Author

Reviewer #2: (No Response)

7. PLOS authors have the option to publish the peer review history of their article (what does this mean?). If published, this will include your full peer review and any attached files.

Reviewer #2: No

---

## [Editor Report · Acceptance letter]

27 Mar 2024

PONE-D-23-29546R1 

PLOS ONE

Dear Dr. Aitken, 

I'm pleased to inform you that your manuscript has been deemed suitable for publication in PLOS ONE. Congratulations! Your manuscript is now being handed over to our production team.

Kind regards, 

on behalf of

Dr. Suresh Yenugu 

Academic Editor

PLOS ONE